# Task-Attentive Transformer Architecture for Continual Learning of Vision-and-Language Tasks Using Knowledge Distillation

**Yuliang Cai**
University of Southern California
caiyulia@usc.edu

**Jesse Thomason**
University of Southern California
jessetho@usc.edu

**Mohammad Rostami**
University of Southern California
rostamim@usc.edu

## Abstract

The size and the computational load of fine-tuning large-scale pre-trained neural networks are becoming two major obstacles in adopting machine learning in many applications. Continual learning (CL) can serve as a remedy through enabling knowledge-transfer across sequentially arriving tasks. However, existing CL algorithms primarily consider learning unimodal vision-only or language-only tasks. We develop a transformer-based CL architecture for learning multimodal vision-and-language (VaL) tasks based on dynamic model expansion and knowledge distillation. Additional parameters are used to specialize the network for each task. Our approach, Task Attentive Multimodal Continual Learning (TAM-CL), enables sharing information between the tasks while addressing catastrophic forgetting. Our approach is scalable, requiring little memory and time overhead. TAM-CL reaches SOTA performance on challenging multimodal tasks. The code is publicly available on https://github.com/YuliangCai2022/TAM-CL.git.

## 1 Introduction

Large-scale pre-trained transformer models are applied in a wide range of applications across modalities, including vision-and-language tasks (Dosovitskiy et al.; Kim et al., 2021; Xu et al., 2023). These models are usually pretrained on a large dataset and then fine-tuned to generalize on a downstream task. Such task-level fine-tuning compromises the model generalizability and necessitates storing a copy of the base model for each task. Continual learning (CL) algorithms (Jin et al., 2021; Yang et al., 2022; Wang et al., 2022; Pelosin et al., 2022; Ermis et al., 2022; Srinivasan et al., 2023) have explored mitigating these challenges for transformers through using a shared model that benefits from cross-task knowledge transfer.

Catastrophic forgetting is the primary challenge in CL (French, 1999). A group of CL algorithms

regularize a fixed shared model to learn each task through different information pathways, i.e., weights (Kirkpatrick et al., 2017; Aljundi et al., 2018). The core idea is to identify a subset of model parameters that are important to encode the learned knowledge about each task and then consolidate these parameters when updating the model to learn new tasks. A second approach is based on model expansion (Rusu et al., 2016; Yoon et al.). The idea is to expand a base model via a small number of additional weights and specialize the network to learn new tasks through these weights. Finally, a group of algorithms use pseudo-rehearsal through experience replay (Rolnick et al., 2019; Mirtaheri et al., 2023). The idea is to store a representative subset of training data for each task in a memory buffer, and replay them back along with the current task's data to maintain the encoded knowledge about the past task. Some methods relax the need for a memory buffer by enabling the model to generate pseudo-samples for the past learned tasks that are used to implement experience replay (Shin et al., 2017; Rostami and Galstyan, 2023). Despite being effective, existing CL methods consider unimodal tasks, e.g., vision-only (Lin et al., 2021; Douillard et al., 2022) or language-only tasks (Jin et al., 2021; Yang et al., 2022), without considering the unique challenges of multimodal tasks, making them inapplicable on VaL tasks.

We develop a new algorithm for learning vision-and-language (VaL) tasks in a CL setting based on dynamic model expansion. To this end, we leverage the self-attention layers of a base bimodal transformer as a shared encoder across all tasks. We then equip the base model with task-attention layers (Douillard et al., 2022) that help to specialize the model for each task using a task-specific token. Our approach requires a small memory overhead and a limited inference time overhead during testing. It also does not need extensive hyper-parameters and remains robust when facing an unknown number

of tasks. Our specific contributions include:

- A dynamically expanding, efficient transformer architecture for multimodal CL.

- A training algorithm to handle diverse, sequentially arriving vision-and-language tasks.

- Extensive experiments to demonstrate that the proposed model achieves SOTA performance.

## 2 Background and Related Work

**Transformers for Vision and Language Tasks:** Multimodal transformers are developed for processing vision-and-language (VaL) tasks (Su et al., 2019; Tan and Bansal, 2019; Kim et al., 2021; Chochlakis et al., 2022; Chen et al., 2020). The core idea is to use modality-specific self-attention layers to extract suitable features from each of the vision and language inputs. These features then are integrated at higher layers to extract cross-modal contextualised representations of the multimodal inputs. The idea is that these global vectors can model the interaction between the vision and the language inputs which is helpful to perform VaL tasks well. The idea has also been adopted in other modalities, including in the context of video processing to relate vision and speech inputs (Arnab et al., 2021; Sun et al., 2019). These transformers are usually trained on a large-scale dataset and have been found to be highly effective when fine-tuned on downstream tasks. Due to a significant performance improvement, transformers are increasingly replacing older architectures.

**Continual learning for transformers:** Despite the successful adoption of transformers in various benchmarks, fine-tuning naturally compromises their generalizability. Using an independent transformer per task would lead to a significant memory load on disk as the size of transformers are increasing dramatically to account for solving more abstract tasks. CL seems to be a natural solution for these challenges but works on CL using transformers are limited. Xin et al. (Jin et al., 2021) use adapters in combination with a hypernetwork to enable CL for language tasks. Alternatively, Yang et al. (Yang et al., 2022) propose a transformer calibration module to make a transformer adaptive. The calibration module is considered to be independent from the base pre-trained transformer and helps to specialize it on a downstream task. The Lifelong Vision Transformer (Wang et al., 2022) utilizes an

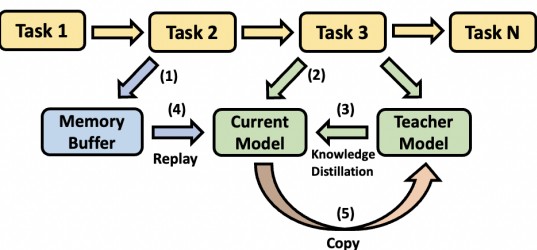

Figure 1: The proposed CL training procedure: (1) A small portion of the data for previous tasks are randomly selected and stored in a memory buffer. (2) The current task arrives with $\mathcal{D}^i$. (3) The training data $\mathcal{D}^i$ is used as input to the teacher model to compute the distillation loss. (4) The memory buffer samples are replayed along with the current task data to train the main model. (5) After learning the current task, the teacher model of the next task will be a copy of the current model.

inter-task attention mechanism to integrate information from previous tasks and reduces the rate at which important attention weights shift away from old tasks towards the current task. Douillard et al. (Douillard et al., 2022) propose a CL architecture for vision tasks using the ViLT model (Kim et al., 2021). Pelosin et al. (Pelosin et al., 2022) extend this work to an exemplar-free setting via distilling the attention-level matrices of transformers to enable model plasticity and to mitigate forgetting effects. Ermis et al. (Ermis et al., 2022) use the idea of adapters in a vision context. To the best of our knowledge, no prior work has explored CL for multimodal tasks using transformer architectures.

## 3 Problem Description

Consider a set of sequentially arriving VaL tasks $\{\mathcal{T}_i\}_{i=1}^T$, each with the annotated training dataset $\mathcal{D}^i = \{\langle (\boldsymbol{I}_i^j, \boldsymbol{L}_i^j)^i, y_i^j \rangle_{j=1}^{N_i}\}$, where $\boldsymbol{I}_i^j \in \mathbb{R}^{H \times W \times C}$ represents the image input, $\boldsymbol{L}_i^j \in \mathbb{R}^{L \times |V|}$ represents the language input, while $y_i^j$ is the text-typed discrete label. The order of these tasks and $T$ are not known a priori. The training data points for $\mathcal{T}_i$ are assumed to be drawn iid from a task-specific joint distribution $p_i^t(\cdot, \cdot, \cdot)$. Our goal in multimodal CL is to learn each task at time-step $i$ and then move forward to learn the next tasks. The learned tasks can be encountered at any time during testing in the future and hence, we would like to maintain the performance of previous learned tasks by preventing catastrophic forgetting.

When learned in isolation, each of these VaL tasks $\mathcal{T}_i$ can be learned using supervised learning conditioned on selecting the suitable predictive model $f_{\theta_M}^i(\cdot, \cdot)$, e.g., a transformer with trainable

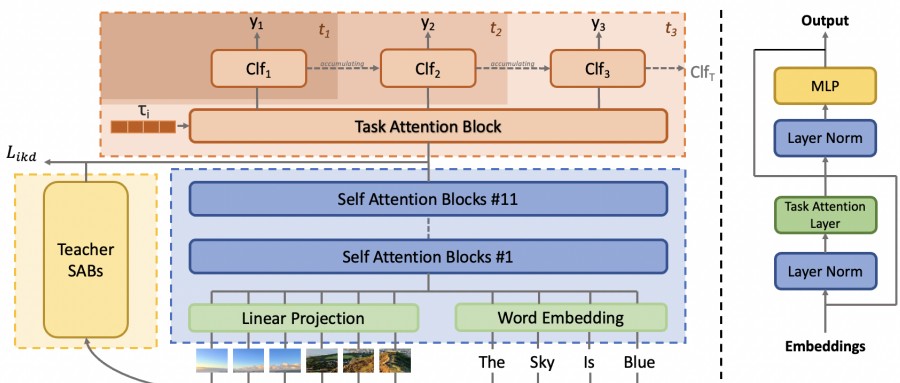

Figure 2: The proposed transformer-based architecture: (left) The VaL inputs are converted into two sequences and then fed into the self-attention layers to generate a fused global feature vector. The data feature vector is then concatenated with the learnable task-specific tokens and then fed into the task attention layer to generate the input for the task-specific classifier heads. The same VaL inputs are also fed into the teacher model's transformer architecture to compute Knowledge Distillation. (right) The task-attention block architecture.

parameters $\theta_M$, and the discrimination loss $\mathcal{L}(\cdot)$, e.g., cross-entropy. However, due to the storage limit, we assume a shared model should be used, as training a separate model per task is impractical. Additionally, only a small portion of training data for each task $\mathcal{T}_i$ can be stored after training for space-saving purpose, which makes multitask learning (Caruana, 1998) an impractical solution. The single task learning strategy using a shared model is not ideal in CL either. Because when the model is updated to learn the current tasks, its performance on the past tasks will degrade due to catastrophic forgetting (Kirkpatrick et al., 2017).

Figure 1 visualizes the high-level presentation of the solution that we propose to address multimodal CL. To use a shared model across all tasks and benefit from knowledge transfer across all tasks, we consider a base transformer model as $f_\theta(\cdot, \cdot)$ and make it adaptive by adding a unique task attention layer after its final layer. Additionally, we modify the loss supervised learning loss by adding a knowledge distillation loss (Hinton et al., 2015) on the intermediate model layers. We consider the teacher model in the knowledge distillation formulation to be a copy of $f_\theta^{i-1}(\cdot, \cdot)$ when training the main student model on $i^{th}$ task $\mathcal{T}_i$. Additionally, we rely on pseudo-rehearsal through experience replay (Rolnick et al., 2019) using a small memory buffer to tackle catastrophic forgetting.

## 4 The Proposed Architecture

Figure 2 visualizes our transformer-based architecture for multimodal CL. The architecture con-

sists of a shared pre-trained replaceable multimodal transformer, an independent task attention block, and MLP classification headers. The task attention block receives task-specific tokens that make the model adaptive. We provide details about these modules and the strategies that we use to train it.

### 4.1 Sequential Feature Extraction Block

We follow the ViLT feature generation procedure (Kim et al., 2021). To build a sequence from the input images, we decompose a given image $I \in \mathbb{R}^{H \times W \times C}$ into patches and then flatten these patches to generate 2D vectors $v \in \mathbb{R}^{N \times (P^2 \cdot C)}$. Here, $C$ is the number of channels, $P \times P$ is the size of each image patch, and $N = HW/P^2$ is the number of patches with size $P \times P$. After generating the set of vectors, we apply a trainable linear projection, $V \in \mathbb{R}^{(P^2 \cdot C) \times H}$, and a position embedding, $V^{pos} \in \mathbb{R}^{(N+1) \times H}$, to transform $v$ into the sequential representation $\overline{v} \in \mathbb{R}^{N \times H}$

$$\overline{v} = [v^{\text{class}}; v_1 V; v_2 V; ...; v_N V] + V^{\text{pos}} \quad (1)$$

We also extract word vectors for $l \in \mathbb{R}^{L \times |V|}$ after applying a word embedding matrix $T \in \mathbb{R}^{|V| \times H}$ and using a position embedding matrix $T^{pos} \in \mathbb{R}^{(L+1) \times H}$, we embed data into $\overline{t} \in \mathbb{R}^{L \times H}$.

$$\overline{t} = [t^{\text{class}}; l_1 T; l_2 T; ...; l_L T] + T^{\text{pos}} \quad (2)$$

We then sum the image and text embeddings with their independent model-type embedding vector $t^{type}$ and $v^{type} \in \mathbb{R}^H$, and concatenate them to build a single sequence $s$.

$$s^0 = [v^{\text{type}} + \overline{v}; t^{\text{type}} + \overline{t}] \quad (3)$$

The combined vector $s$ is passed through $D$ classic self-attention layers of the base VaL transformer, i.e., ViLT, and then are fed into the task attention layer, with the output embedding from the transformer attention layers denoted as $s^D$ and output from the task attention layer as $s^{D+1}$.

$$\hat{s}^d = MSA(LN(s^{d-1})) + s^{d-1}, d = 1, ..., D$$
$$s^d = MLP(LN(\hat{s}^d)) + \hat{s}^d, d = 1, ..., D \quad (4)$$

## 4.2 Task Attention Block

The core idea of our work lies in adopting the idea of self-attention on task-level, each time a new task is learned. Different from the vanilla input data level self-attention layer, the task-attention layer is a task-focused attention layer that a trainable task token is initialized for each new task, denoted as $\tau_i \in \mathbb{R}^{G \times 1}$ for task $i \subseteq [1, 2, .., T]$, where $G$ is the size of latent space of each self-attention layer. Similar to the self-attention block (SAB), the task attention block (TAB) is a module which consists of an attention layer, layer normalization, and MLP. The attention layer is instantiated as a task-attention layer, rather than vanilla self-attention. Task attention block takes two inputs, the output of the self-attention blocks $s^D$ and the task token $\tau$, note that the same task token $\tau$ is used for all the input instances in that task-specific train/test dataset. The two vectors are concatenated to generate an input for task attention:

$$s'^{D+1}_i = [\tau_i, s^D] \in \mathbb{R}^{(N+1) \times G}, i = 1, ..., T$$
$$\hat{s}^{D+1}_i = TA(LN(s'^{D+1}_i)), i = 1, ..., T$$
$$s^{D+1}_i = MLP(LN(\hat{s}^{D+1}_i)) + \hat{s}^{D+1}_i, i = 1, ..., T \quad (5)$$

The task attention block is placed after the last self-attention block of the transformer. While we can have more than one task attention block, our architecture uses a single TAB. The operation of the task attention layer is given as follows:

$$Q_i = W_q \times \tau_i,$$
$$K_i = W_k \times s^{D+1}_i,$$
$$V_i = W_v \times s^{D+1}_i, \quad (6)$$
$$A_i = Softmax(Q_i \cdot K_i^T / \sqrt{G/h}),$$
$$O_i = W_o A_i V_i + b_o \in \mathbb{R}^{1 \times G}$$

where $h$ is the number of attention heads in the transformer (Vaswani et al., 2017).

Finally, the output of the task attention block, $s^{D+1}_i$ is then fed into task-specific classifier layers:

$$y_i = Clf_i(s^{D+1}_i), i = 1, ..., T \quad (7)$$

# 5 Training Algorithm

The architecture in Figure 2 visualizes a snapshot of our model, TAM-CL, at a given timestep. Note that the architecture is dynamically expanded as more tasks are learned sequentially. We describe the suitable CL loss functions that we use for training.

## 5.1 Token Expansion

During the training stage, the transformer's self attention blocks and the task attention block are shared among all the tasks. However, for each of the new tasks, we define a new task token with the same dimension, $\tau \in G \times 1$, and initialize a new accumulative task-specific classifier, $Clf_i(\cdot)$, for task $i$, which the output dimension of $Clf_i(\cdot)$ is expanded based on $Clf_{i-1}(\cdot)$. With more tasks added on, the output dimension of task-specific classifier $i$ would be accumulating in the way:

$$E_i = E_i^{orig} + E_{i-1}, i = 1, ..., T, \quad (8)$$

Where $E_i$ denotes the output dimension for $i^{th}$ classifier, $E_i^{orig}$ is the output dimension for $i^{th}$ task in its original design. For task $i$, we combine the $i^{th}$ task token with the updated path token, $s^D$ from the last self-attention block of the transformer, and send it into the task attention block, as described in Sec 4.2. At this stage, only the $i^{th}$ task token and task-specific classifier would be trainable, while all the other task tokens and classifiers remain frozen.

During the testing stage, the task number, $i$, of test data is explicitly given, and $s^D$ is combined with the $i^{th}$ learned task token to feed into task attention block along with using its corresponding task-specific classifier, while all other the task tokens and classifiers remain unused.

## 5.2 Loss and Knowledge Distillation

Our objective function consists of three loss terms: (i) cross-entropy Loss, $\mathcal{L}_c$, which is the original objective function for each task in single task learning setting. Note it can vary from task to task, (ii) the knowledge distillation (KD) loss, which is computed from $s^D$ of the main student model and the teacher model, $\mathcal{L}_{ikd}$, (iii) the diverse loss $L_{div}$ which compares the data distribution of task tokens and makes them more diverse. The final loss is:

$$\mathcal{L} = (1 - \lambda)\mathcal{L}_c + \lambda\alpha\mathcal{L}_{ikd} + \beta\mathcal{L}_{div}, \quad (9)$$

where $\lambda$ is set to $\frac{T_n-1}{T_n}$, $T_n$ denotes the total number of tasks we have seen so far, $\alpha$ denotes a constant which varies for different tasks, and $\beta = min(\mathcal{L}_{div}, 0.1 \times ((1-\lambda)\mathcal{L}_c + \lambda\alpha\mathcal{L}_{ikd}))$.

The application of intermediate knowledge distillation is the core of TAM-CL, which aims to distill the knowledge from the teacher model into the main student model in order to constrain the distribution shift and prevent catastrophic forgetting. In our architecture, the teacher model is a copy of model $f_\theta^{i-1}(\cdot, \cdot)$ when training on $i^{th}$ task. Different from most other methods which apply knowledge distillation by computing the loss from the last layer output, $y$ and $y_{teacher}$, we introduce an intermediate knowledge distillation loss, i.e., the loss term $\mathcal{L}_{ikd}$ is computed between the last self-attention block of the transformer and the task attention block, $s^D$. Through experiments in Section 6.2, we find that compared with knowledge distillation computed from the last layer output, eg. Dytox, $y$ and $y_{teacher}$, such an intermediate knowledge distillation is helpful in the architecture of a pre-trained model followed by a non-pre-trained block, which could constrain the probability shift of pre-trained parameters and leave the rest of the layers flexible enough to learn new tasks. To our best knowledge, TAM-CL is the first to introduce intermediate knowledge distillation objective function in multimodal continual learning.

## 5.3 Experience Replay

The above training procedure enables training a shared model across the tasks but still, it need datasets for all tasks which breaks a primary assumption in CL. To address this issue, we use a memory buffer during the training stage at each time-step which stores a tiny percentage, e.g., $\approx 5\%$, of the training dataset for all the previous tasks. When learning the current task with a specific batch number, the next batch will be randomly selected from the memory buffer to consolidate the parameter distribution on previous tasks. As a result, forgetting effects will be mitigated.

We introduce the Task Attentive Multimodal Continual Learning (TAM-CL) training procedure, which enables stable and high-performances while mitigating forgetting effects in Multimodal continual learning in Algorithms 1 and 2.

## 6 Experimental Results

We evaluate TAM-CL using five VaL tasks.

---

**Algorithm 1** TAM-CL Train

**INPUT:** Model **M**, MemBuffer **B**, ReplayFreq **f**
**for** epoch in num_epoch **do**
    **for** step, batch in dataloader **do**
        Loss $\leftarrow$ TrainStep(**M**, batch)
        **if** step % **f** == 0 **then**
            batch$_{replay}$ $\leftarrow$ **B**.getBatch()
            Loss$_{replay}$ $\leftarrow$ TrainStep(**M**, batch$_{replay}$)

---

**Algorithm 2** TAM-CL TrainStep

**INPUT:** Model **M**, Teacher Model **T**, Batch, Target **t**, Token **k**
p $\leftarrow$ Model(Batch)
$loss \leftarrow$ CrossEntropyLoss(p, **t**)
ModelSabPre $\leftarrow$ **M**.SAB(Batch)
TeacherSabPre $\leftarrow$ **T**.SABs(Batch)
$loss_{ikd} \leftarrow$ KL-Div(ModelSabPre, TeacherSabPre)
$loss_{div} \leftarrow$ CrossEntropyLoss($\mathbf{k}_i, \mathbf{k}_j$)   $j = 1, .., i-1$
$loss \leftarrow (1-\lambda)loss + \lambda\alpha loss_{ikd} + \beta loss_{div}$
$loss$.backward()
**Return** loss

---

### 6.1 Experiment Setup

We use five VaL datasets to generate sequential tasks. We use **SNLI-VE** (Xie et al., 2019), an image-sentence pairs dataset whereby a premise is defined by an image, rather than a natural language sentence as in traditional Textual Entailment tasks, **COCOQA** (Ren et al., 2015), a visual question answering dataset based on Microsoft COCO image dataset, **GQA** (Hudson and Manning, 2019), a compositional question-answering and visual reasoning dataset leverages scene graph structure, **NLVR2** (Suhr et al., 2018), a visual reasoning dataset which takes two images and determine the correctness of the given sentence, **OKVQA** (Marino et al., 2019), a knowledge-based visual question-answering dataset whhere the image content is not sufficient to answer questions. Due to the computational limits, we trained all the models on part of the whole dataset, where the maximum size of the training examples are 80000. Table 1 provides statistics of these dataset.

We use ViLT model with pre-trained weights, "BERT-base-uncased" in our experiments. To main-

| Name | # Training Examples | # Labels |
|---|---|---|
| NLVR2 | 80000 | 2 |
| SNLI-VE | 80000 | 3 |
| COCOQA | 78736 | 430 |
| GQA | 80000 | 1842 |
| OKVQA | 18032 | 2910 |

Table 1: Statistics of the VaL dataset.

| COCOQA → NLVR2 → OKVQA → SNLI-VE → GQA | | | | | |
|---|---|---|---|---|---|
| | COCOQA | NLVR2 | OKVQA | SNLI-VE | GQA |
| **TAM-CL** | **66.09 (13.15%)** | **66.07 (14.87%)** | **21.24 (22.59%)** | **64.05 (19.13%)** | 50.86 |
| Finetune | 40.67 (47.09%) | 53.85 (79.42%) | 8.26 (74.37%) | 53.83 (46.81%) | **51.92** |
| FDR | 48.74 (32.12%) | 55.91 (29.89%) | 11.59 (55.44%) | 59.30 (28.26%) | 50.67 |
| EWC | 51.44 (33.10%) | 60.87 (42.19%) | 16.16 (49.40%) | 57.93 (35.21%) | 49.67 |
| ER | 56.30 (27.20%) | 62.06 (34.27%) | 15.62 (50.10%) | 61.12 (27.91%) | 50.12 |
| Dytox | 60.67 (20.52%) | 65.56 (15.37%) | 10.41 (25.26%) | 62.26 (20.70%) | 11.12 |
| Avg. | 53.98 (28.63%) | 60.72 (36.00%) | 13.88 (46.19%) | 59.75 (29.67%) | 44.06 |

Table 2: Comparative: the accuracy and forgetting rate for each task in two different task sequences. For each task sequence, each value means the **final accuracy** of that task after learning the last task, and **(forgetting rate %)** means the forgetting rate of final accuracy compared with the best accuracy of each task. The last row represent the average accuracy and forgetting rate of each task.

| OKVQA → GQA → COCOQA → SNLI-VE → NLVR2 | | | | | |
|---|---|---|---|---|---|
| | OKVQA | GQA | COCOQA | SNLI-VE | NLVR2 |
| TAM-CL | **15.09 (54.56%)** | **41.91 (27.59%)** | **60.86 (18.36%)** | **60.47 (29.34%)** | **65.86** |
| ablation TAB | 13.97 (57.22%) | 39.95 (28.69%) | 59.92 (19.65%) | 57.35 (37.22%) | 65.71 |
| ablation $\mathcal{L}_{ikd}$ | 11.53 (65.36%) | 37.19 (34.07%) | 58.62 (21.71%) | 58.34 (35.30%) | 65.21 |
| ablation replay | 2.11 (93.69%) | 27.25 (51.84%) | 41.59 (44.33%) | 50.06 (56.40%) | 65.64 |

Table 3: Ablation: the accuracy and forgetting rate for each task in ablation experiments.

tain consistency in our comparison, we use ViLT with the same pre-trained parameters for all the experiments. Hence, we have 11 self-attention blocks (SAB), with dimension of 768 and attention heads of 12. We then attach one task attention block (TAB) after the transformer encoder which also has 768 hidden dimensions and 12 attention heads. For all four tasks, we apply AdamW optimizer with $l = $ 1e-2, $\epsilon = $ 1e-8, $\beta_1 = 0.9$, and $\beta_2 = 0.98$.

Since there is no prior method for multimodal CL, we use extensions of **Dytox** (Douillard et al., 2022), **FDR** (Titsias et al., 2020), **EWC** (Kirkpatrick et al., 2017), **Experiment Replay** (Rolnick et al., 2019), and **Direct Fine-Tuning**, as five alternatives for comparison. Note that direct fine-tuning serves as a lowerbound to measure the effect of catastrophic forgetting and effectiveness of CL.

After learning each task $\mathcal{T}_i$, we evaluate the forgetting rate on previous tasks $T_k$, $k \in 1, .., i-1$. To study the effect of task order, we perform experiments with different task orders that reflect the difficulty of tasks. For each of the learned tasks, the final performance accuracy and the forgetting rates compared with its best accuracy are reported. To assist the analysis of task order's impact to the experiment's performance, we intuitively

rank the difficulty level of the five tasks by a metric: $\frac{\# \ Training \ Examples}{\# \ Labels}$, which are presented in Table 1. Roughly, the bigger this value is, the easier the tasks is. In this case, we rank our five tasks from difficult to easy as: **OKVQA**, **GQA**, **COCOQA**, **SNLI-VE** and **NLVR2**, of which corresponding scores are 6.19, 43.43, 183.10, 26666.66 and 40000. To eliminate the bias due to a specific task order, we perform experiments on several task orders. More results are included in the Appendix.

Most CL algorithms consider relatively homogeneous tasks. To evaluate the algorithm capacity for preventing catastrophic forgetting on heterogeneous tasks that we have, we use the following normalization-based metric (Srinivasan et al.):

$$\mathbb{T}_{\mathbf{F}}(j \leftarrow i) = \frac{S_A^j - S_A^{j \leftarrow i}}{S_A^j - S_R^j} \quad (10)$$

where $\mathbb{T}_{\mathbf{F}}(j \leftarrow i)$ stands for the forgetting rate of task j after learning task i, $S_A^j$ denotes the accuracy of task $j$ before learning new tasks, $S_A^{j \leftarrow i}$ denotes the accuracy of task $j$ after learning task $i$, $i > j$, and $S_R^j$ means the accuracy of task $j$ by randomly choosing the output label, which is calculated by $\frac{1}{\# \ labels}$. In other words, Eq. (10) enables comparing forgetting rates across tasks that are consider-

ably different because we are measuring how well the model is preforming compared to a baseline of total forgetting for that task. More details about experimental setup are included in the Appendix.

## 6.2 Comparative Results

We report our performance results after training on all five tasks in Table 1. As expected, we observe that fine-tuning is the worst algorithm regarding the forgetting rates for all tasks before GQA. On the other hand, the accuracy of GQA is the highest among all the methods. The high accuracy is also expected as there is a trade-off between accuracy and forgetting rate of a task, without any constrain to the forgetting rate, the accuracy of fine-tuning method is supposed to be relatively higher. This empirical observation demonstrates the significance of adopting CL algorithm for learning multimodal tasks sequentially. We adopt two regularization-based CL methods, EWC and Function Distance Regularization (FDR), in our comparative experiments, and observe that most of results are below the average accuracy for all the previous tasks. EWC and FDR often is effective when used on smaller models but in line with prior observations in the case of using transformers for unimodal tasks (Srinivasan et al.; Jin et al., 2021; Douillard et al., 2022), we conclude that regularization-based CL methods are not suitable methods for CL with large transformers. This result suggests that transformers are sensitive with respect to weight consolidation because their learning capacities are compromised significantly when many weights are frozen.

In contrast, the accuracy and forgetting rate results for experience replay method is more decent. In Table 2, all the accuracy results of experiment replay are above the average line, which indicates that experiment replay is a stable and relatively efficient CL method which is not constrained to only small models and unimodal data. We also adopt more state-of-the-art method, Dytox, as our baseline. However, we observe that although Dytox has some result close to those of our method in some cases, the rest of the accuracies are relatively lower compared with other methods. For example, Dytox's accuracy on OKVQA is 10.41, only higher than fine-tuning. Noticing that this accuracy is corresponded with a low forgetting rate, 25.26%, which indicates that Dytox prevents forgetting by underfitting the current task. Similarly, the accuracy of GQA is 11.12, which verifies that Dytox's

underfitting on current task. While Dytox is designed for unimodal tasks, we conclude that Dytox is not suitable for multi-modal tasks.

Finally, TAM-CL outperforms every other method in Table 2. Especially, it is 8.93% higher than the second best method for COCOQA, 31.43% higher than the second best method for OKVQA, and only 2.08% lower than the best accuracy for the last task, GQA. In the cross-task, multimodal CL scenario, TAM-CL is capable for significantly reducing the forgetting rate of previous tasks while maintaining the high accuracy of current task. Additional experiments are included the Appendix.

## 6.3 Ablation Results

To reflect the necessity of each component in our design, ablation experiments are performed on the effect of $\mathcal{L}_{ikd}$ loss function, the training strategy with experience replay, and using the task attention block, respectively. In ablation experiments, we use the task sequence order OKVQA $\rightarrow$ GQA $\rightarrow$ COCOQA $\rightarrow$ SNLI-VE $\rightarrow$ NLVR2.

In the ablative experiment, we choose the full TAM-CL pipeline as the baseline and compare the performance of TAM-CL to each ablation task. Table 3 presents results for our ablation experiments. We observe that the full pipeline for TAM-CL leads to the best score in terms of both the forgetting rate and the performance accuracy. These results validate the necessity of every component in our approach for an optimal performance. We observe that the effect of dropping the $\mathcal{L}_{ikd}$ loss leads to an average performance drop of 12.74% across the four tasks, which demonstrates the significance of intermediate knowledge distillation to CL.

Meanwhile, we also observe that when ablating the whole task-attention block, which also includes the $\mathcal{L}_{ikd}$ loss, the forgetting rates and accuracy performances are slightly better than only ablating the $\mathcal{L}_{ikd}$ loss. This higher performance may look unintuitive but our hypothesis is that the loss term $\mathcal{L}_{ikd}$ is specifically important to train the task-attention layer which is the main continual learning component in the model, leading to being more impactful.

As expected, our training strategy using experience replay is also contributing to the optimal performance by mitigating catastrophic forgetting of previous tasks. We observe that experience replay is more helpful maintaining the accuracy of the early tasks in the sequence. For example, in Table 3, by ablating experience replay, the forgetting rate

| GQA → COCQA → OKVQA → SNLI-VE → NLVR2 | | | | | | | | | |
|---|---|---|---|---|---|---|---|---|---|
| COCOQA | OKVQA | | SNLI-VE | | | NLVR2 | | | |
| G | G | C | G | C | O | G | C | O | S |
| 3.22% | 6.81% | 5.81% | 4.60% | 4.58% | 8.51% | 9.43% | 9.72% | 20.72% | 14.98% |
| COCOQA → NLVR2 → SNLI-VE → OKVQA → GQA | | | | | | | | | |
| NLVR2 | SNLI-VE | | OKVQA | | | GQA | | | |
| C | C | N | C | N | S | C | N | S | O |
| 7.40% | 8.45% | 6.58% | 11.08% | 14.57% | 19.23% | 19.30% | 23.34% | 24.98% | 27.29% |

Table 4: Task order: the forgetting rate for each task in two different task sequences. The letter on each task sequence represent the shortcut of the task name: **S**:SNLI-VE, **N**:NLVR2, **C**:COCOQA, **O**:OKVQA, **G**:GQA. The second row of each task sequences represents the current task, while the third row of each sequence represent the previous task. Eg. After trained on COCOQA, the forgetting rate of G(QA) is 15.42%.

of OKVQA after learning NLVR2 is 93.69%, while the forgetting rate of GQA after learning NLVR2 is 51.84%, and the forgetting rate of COCOQA after NLVR2 is 44.33%, which are significantly less than 93.69%. These results demonstrate that the optimal performance of our method stems from using all the three primary ideas that we proposed.

## 6.4 Effect of Different Task Orders

To further analyze the performance of TAM-CL, we compare the performance of TAM-CL on two different task sequences and analyze the impact of the task order on catastrophic forgetting. Ideally we would like to develop an algorithm that works well on all task orders. Although in practice we don't control the task order and the tasks are encountered in an order determined by the environment, we study the effect of task order, assuming it is given.

Table 4 presents the forgetting rate results for every time step. In another word, we evaluate the forgetting rate of previous tasks after the training of every single task. Inspecting results from both sequence, we conclude that the task order is a key factor to the performance of a certain task. For example, in sequence 1, OKVQA is the third task, and the forgetting rate of OKVQA is 20.72% after learning the last task. Meanwhile, in sequence 2, OKVQA is the fourth task, but its forgetting rate is 27.29%, which is higher than that of the previous sequence. By our intuitive metric of task difficulty described in 6.1, our hypothesis is that, in sequence 1, SNLI-VE and NLVR2 are two easier tasks, which requires less parameter distribution shift to achieve high accuracy, thus the parameter distribution for OKVQA is less shifted. However, in sequence 2, GQA is the second difficult task,

which requires more distribution shift from the previous task OKVQA, to achieve higher performance. Such a correlations between the forgetting rate and task difficulty can also be verified in sequence 2. For example, the forgetting rate of COCOQA after training the two relatively easier tasks, NLVR2 and SNLI-VE, are 7.40% and 8.45%. In contrast, after training OKVQA and GQA, the forgetting rate of OKVQA is 11.08% and 19.30%, which are much higher than after NLVR2 and SNLI-VE. For additional experiments, please refer to the Appendix.

We also observe that even though the NLVR2 is a relatively easier task, in sequence 1, after trained on NLVR2, all the forgetting rate of previous tasks rise more than twice. Our hypothesis is that altough NLVR2 is easy to train, it has an essential difference from other tasks that NLVR2 takes two images as a single input and perform visual reasoning while all the other tasks only take one image at a time. Such a unique property of NLVR2 task raises the extent of parameter distribution shift, but not as drastically as the difficulty of task does.

Meanwhile, we surprisingly find that after trained on some specific tasks, the forgetting rate of previous tasks can even decrease. In sequence 1, after trained on OKVQA, the forgetting rate of GQA and COCOQA are 6.81% and 5.81% respectively. However, after trained on the next task, SNLI-VE, the forgetting rates of those tasks drop to 4.60% and 4.58%, which might indicate the potential forward transfer capacity that we can further explore.

## 7 Conclusions

We developed an algorithm for multimodal continual learning for transformer architectures based on dynamic model expansion and knowledge distilla-

tion. We use a task-attention block which specializes the transformer architecture for a particular tasks using a special learnable task token. Knowledge distillation helps to benefit from knowledge transfer across the tasks for positive forward transfer. We mitigate catastrophic forgetting using experience replay. Our experiments demonstrate that our approach is effective and leads to state-of-the-art performance in terms of both forward transfer and catastrophic forgetting. Our TAM-CL architecture is a first algorithm in studying CL in multimodal settings and demonstrates that more explorations in this direction is necessary.

## Limitations

Although TAM-CL reaches the state-of-the-art performance, it has its own limitations and we anticipate several future research direction to address these limitation. More specifically:

- As we have three visual question answering tasks, one visual reasoning task and one visual entailment task, the designed experiments are not only cross-task, which is desired, but also cross-domain. We will further explore the performance of TAM-CL in single domain cross-task settings and compare with the other state-of-the-art methods.

- In multimodal learning scenario, the modal can not only take multimodal input, but also uni-modal input, by setting the input of other modalities to some constant number. We will further explore TAM-CL's capacity on uni-modal tasks and compare the result with other state-of-the-art methods.

- Due to the computational limits, some of the task that we trained on are not in its full version. For example, we are training TAM-CL on 80000 training examples for SNLI-VE dataset, where the full SNLI-VE dataset contains 529527 training examples. We will further explore the performance of TAM-CL on the full size of training examples for all the tasks.

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

# A    Appendix

## A.1    Implemntation Details

### A.1.1    Data Preprocessing

As the tasks, which the model trains on, have completely different dataset, we compress the image from its original size to (384, 640) before sending to the linear projection embedding block. In the self-attention layers of the pre-trained transformer and task-attention block, the image-text feature vector has the shape of (batch, 768).

Different from tasks such as COCOQA, PathVqa, and SNLI-VE which takes only 1 input image, NLVR2 takes 2 input images with a hypothesis as text input. During the training stage, the text input is combined with one image every time before feeding into the transformer, and concatenate the output from two images as one. Consequently, the output from transformer will be (batch, 1536). However, due to the input size limitation of the task-attention block, we compress the vector, $\mathbf{V}$, from 1536 to 768 by taking the average value of the adjacent element:

$$\mathbf{V}'[i] = \frac{\mathbf{V}[i] + \mathbf{V}[i+1]}{2}, i = 0, 2, 4, ..., 1534$$
(11)

, where $\mathbf{V}'$ is the compressed feature vector. We then feed $\mathbf{V}'$ into the task-attention block.

### A.1.2    Hyperparameters for the Experiments

For all experiments we perform, we use single A100 GPU with batch size of 16.

For the training epochs, due to the limit of computational force, we only train each task with small epochs: 5 epochs for SNLI-VE, 10 epochs for CO-COQA, GQA and NLVR2, 20 epochs for OKVQA.

For EWC method, we applied the fisher sample percentage as 0.4, which means 40% of dataset are collected to build the fisher matrix. During the training stage, we set the EWC weight as 0.1. For Experience Replay method, the sample size of data in memory buffer is 5%, and the sample frequency is 100, which means for every tasks, we randomly extract 5% of data and store them into the memory buffer. During the training stage, after 100 batches of current dataset are trained, we randomly select a batch of data from memory buffer and train the model on that batch from a random previous task. For TAM-CL, as the knowledge distillation loss is extremely small compared with other loss, we set the weight for $\mathcal{L}_{ikd}$ as 5000 for all the four tasks. As in TAM-CL, we adopt experience replay as our

training strategy, we also set the sample size of memory size as 5%, and the sample frequency as 100.

## A.2    Results on Additional Task Orders

Due to space limit, additional comparative result on different task sequence of all time steps are presented here and each table presents the accuracy of the task right after trained on it and the forgetting rate of previous tasks after trained on the current task. Through the four tables, we observe that TAM-CL has the leading forgetting rate in different time steps in most of the cases.

As discussed in Section 6.2, the two regularization-based method, EWC and FDR are not suitable for large-model and long sequence continual learning. We observe that in some cases, the forgetting rate of FDR and EWC are close or even higher than the finetuning method, which means in such condition, those methods are not capable for preventing catastrophic forgetting. For example, in Table 6, after trained OKVQA, the forgetting rate of NLVR2 for Finetune method is 59.13%, while that of EWC is 52.07%, which is close to the performance of without CL algorithm. Meanwhile, after trained on GQA, the forgetting rate of OKVQA for Finetune method is 40.90% and that of EWC is 42.28%, which is even higher than the non-CL method baseline. In table 8, after trained on NLVR2, the forgetting rate of SNLI-VE for Finetune is 50.07, and forgetting rate for FDR is 47.51, for EWC is 54.04%, which are close to and above the non-CL baseline.

Regarding the experience replay method, it provides relatively high accuracy and low forgetting rates compared to the two regularization methods. However, the capacity of ER for preventing catastrophic forgetting in later tasks are not as stable as it is in early tasks. For example, in Table 5, after trained on the last task, GQA, the forgetting rate of OKVQA for ER is 50.10%, which is close to FDR and higher than EWC. Comparatively, after trained on SNLI-VE, the forgetting rate of OKVQA for ER is 26.57%, far greater than the forgetting rate for FDR, which is 52.28%. In contrast, Dytox method has the stable capacity to prevent catastrophic forgetting, which is not affected drastically by length of task sequence, however, it lost the competitiveness due to its low performance on training the current task. We observe that in Table 5, the accuracy of OKVQA for Dytox is 19.12, whereas the

**COCOQA → NLVR2 → OKVQA → SNLI-VE → GQA**

| | COCOQA | NLVR2 | | OKVQA | | |
| --- | --- | --- | --- | --- | --- | --- |
| | | | COCOQA | | COCOQA | NLVR2 |
| TAM-CL | 76.09 | **68.88** | **7.89%** | 31.63 | **9.45%** | **3.72%** |
| Finetune | 75.88 | 68.71 | 70.21% | **32.24** | 19.41% | 45.75% |
| EWC | 75.89 | 67.93 | 73.65% | 31.39 | 20.95% | 64.20% |
| FDR | 71.81 | 58.43 | 28.99% | 26.01 | 24.54% | 59.07% |
| ER | **76.77** | 68.35 | 12.08% | 31.19 | 13.64% | 34.04% |
| Dytox | 76.68 | 68.41 | 10.37% | 13.93 | 16.15% | 23.35% |

| | SNLI-VE | | | |
| --- | --- | --- | --- | --- |
| | | COCOQA | NLVR2 | OKVQA |
| TAM-CL | 71.32 | **10.25%** | **6.98%** | 15.19% |
| Finetune | 71.6 | 31.71% | 41.57% | 71.71% |
| EWC | 71.03 | 23.55% | 43.49% | 29.75% |
| FDR | 68.56 | 71.13% | 51.00% | 52.28% |
| ER | **71.37** | 12.05% | 10.46% | 26.57% |
| Dytox | 69.82 | 14.30% | 16.25% | **2.15%** |

| | GQA | | | | |
| --- | --- | --- | --- | --- | --- |
| | | COCOQA | NLVR2 | OKVQA | SNLI-VE |
| TAM-CL | 50.86 | **13.15%** | **14.87%** | **22.59%** | **19.13%** |
| Finetune | **51.92** | 47.09% | 79.42% | 74.37% | 46.81% |
| EWC | 49.67 | 33.10% | 42.19% | 49.40% | 20.45% |
| FDR | 50.67 | 32.12% | 29.89% | 55.44% | 28.26% |
| ER | 50.12 | 27.20% | 34.27% | 50.10% | 27.19% |
| Dytox | 11.12 | 20.52% | 15.37% | 25.26% | 20.70% |

Table 5: Accuracy and forgetting rate of task order: COCOQA → NLVR2 → OKVQA → SNLI-VE → GQA

accuracy for all the other methods are above 30. Meanwhile, Dytox only obtains 6.10 in GQA and the accuracy for all other methods are above 50.

Finally, although TAM-CL is not leading on every single forgetting rate, it outperforms the rest of methods on 87.5% of the total forgetting rates. For the rest of the exceptional cases, TAM-CL has the above-average performance which proves its capacity and stability to prevent catastrophic forgetting.

We also aware that the accuracy of TAM-CL is not always leading among the six methods. However, as all of the differences between the top accuracy and TAM-CL's accuracy are below 5%, and as our main focus is on the improvement of forgetting rate, we consider the slightly accuracy difference is in an acceptable range.

**COCOQA → NLVR2 → SNLI-VE → OKVQA → GQA**

| | COCOQA | NLVR2 | | SNLI-VE | | |
| --- | --- | --- | --- | --- | --- | --- |
| | | | COCOQA | | COCOQA | NLVR2 |
| TAM-CL | 76.73 | 69.99 | **7.40%** | 71.12 | **8.45%** | **6.58%** |
| Finetune | **76.95** | 67.83 | 74.75% | **71.72** | 62.51% | 38.12% |
| EWC | 76.69 | **71.67** | 64.88% | 70.84 | 58.61% | 27.77% |
| FDR | 72.43 | 57.99 | 31.31% | 67.59 | 38.96% | 65.08% |
| ER | **76.38** | 70.35 | 8.15% | 70.69 | 8.65% | 8.52% |
| Dytox | 76.63 | 68.89 | 8.50% | 70.45 | 9.43% | 7.94% |

| | OKVQA | | | |
| --- | --- | --- | --- | --- |
| | | COCOQA | NLVR2 | SNLI-VE |
| TAM-CL | 30.47 | **11.08%** | **14.57%** | 19.23% |
| Finetune | **32.12** | 19.19% | 59.13% | 22.22% |
| EWC | 31.68 | 18.83% | 52.07% | **15.44%** |
| FDR | 11.49 | 44.23% | 45.52% | 41.27% |
| ER | 30.91 | 12.92% | 28.12% | 17.92% |
| Dytox | 19.12 | 17.16% | 32.55% | 37.06% |

| | GQA | | | | |
| --- | --- | --- | --- | --- | --- |
| | | COCOQA | NLVR2 | SNLI-VE | OKVQA |
| TAM-CL | 56.20 | **19.30%** | **23.34%** | **24.98%** | **27.29%** |
| Finetune | 57.03 | 36.53% | 66.16% | 34.11% | 40.90% |
| EWC | **57.21** | 33.51% | 60.60% | 32.84% | 42.28% |
| FDR | 42.67 | 41.08% | 46.41% | 74.08% | 50.17% |
| ER | 57.04 | 26.04% | 32.89% | 27.97% | 41.33% |
| Dytox | 6.10 | 21.81% | 38.50% | 30.10% | 34.43% |

Table 6: Accuracy and forgetting rate of task order: COCOQA → NLVR2 → SNLI-VE → OKVQA → GQA

| GQA → COCOQA → OKVQA → SNLI-VE → NLVR2 | | | | | | |
|---|---|---|---|---|---|---|
| | GQA | COCOQA | | OKVQA | | |
| | | | GQA | | GQA | COCOQA |
| TAM-CL | 56.58 | 74.44 | 3.22% | 30.44 | **6.81%** | 5.81% |
| Finetune | 58.11 | 75.95 | 10.60% | **31.99** | 12.78% | 10.65% |
| EWC | **58.69** | **76.12** | 10.45% | 31.15 | 13.46% | 7.97% |
| FDR | 55.50 | 71.42 | 12.70% | 28.39 | 12.17% | 6.21% |
| ER | 57.61 | 75.17 | 4.17% | 31.21 | 9.58% | **4.26%** |
| Dytox | 56.83 | 75.22 | **1.93%** | 29.87 | 11.29% | 7.87% |

| SNLI-VE | | | |
|---|---|---|---|
| | GQA | COCOQA | OKVQA |
| TAM-CL | 71.49 | **4.60%** | **4.58%** | **8.51%** |
| Finetune | **72.54** | 9.91% | 7.35% | 15.93% |
| EWC | 72.10 | 10.82% | 6.43% | 11.46% |
| FDR | 70.17 | 8.29% | 7.11% | 13.43% |
| ER | 72.39 | 6.00% | 5.03% | 10.47% |
| Dytox | 72.44 | 8.13% | 5.40% | 10.07% |

| NLVR2 | | | | |
|---|---|---|---|---|
| | GQA | COCOQA | OKVQA | SNLI-VE |
| TAM-CL | 70.04 | **9.43%** | **9.72%** | **20.72%** | **14.98%** |
| Finetune | 70.32 | 55.09% | 63.90% | 77.80% | 53.18% |
| EWC | 67.57 | 57.47% | 63.77% | 80.89% | 51.90% |
| FDR | 53.60 | 33.54% | 56.20% | 63.64% | 36.42% |
| ER | **70.38** | 13.19% | 9.90% | 23.43% | 19.93% |
| Dytox | 67.34 | 15.86% | 13.90% | 41.45% | 27.61% |

Table 7: Accuracy and forgetting rate of task order: GQA → COCOQA → OKVQA → SNLI-VE → NLVR2

| COCOQA → GQA → SNLI-VE → OKVQA → NLVR2 | | | | | | |
|---|---|---|---|---|---|---|
| | COCOQA | GQA | | | SNLI-VE | |
| | | | COCOQA | | COCOQA | GQA |
| TAM-CL | 76.59 | 57.83 | **3.86%** | 72.03 | **3.37%** | **1.66%** |
| Finetune | 76.84 | **58.72** | 18.86% | 72.09 | 23.57% | 5.64% |
| EWC | 76.49 | 57.27 | 18.65% | 72.11 | 21.18% | 4.26% |
| FDR | 72.00 | 53.42 | 14.66% | 68.25 | 20.47% | 2.74% |
| ER | **76.88** | 58.07 | 17.52% | 72.19 | 9.01% | 1.71% |
| Dytox | 76.77 | 25.41 | 4.51% | **72.33** | 3.77% | 2.14% |

| | OKVQA | | | |
|---|---|---|---|---|
| | | COCOQA | GQA | SNLI-VE |
| TAM-CL | 30.74 | **6.64%** | **4.78%** | 5.66% |
| Finetune | 32.77 | 24.37% | 9.89% | 9.27% |
| EWC | 31.78 | 22.88% | 5.88% | 8.65% |
| FDR | 26.57 | 26.88% | 8.84% | **4.77%** |
| ER | **33.22** | 11.64% | 6.11% | 5.74% |
| Dytox | 13.74 | 10.61% | 4.98% | 6.74% |

| | NLVR2 | | | | |
|---|---|---|---|---|---|
| | | COCOQA | GQA | SNLI-VE | OKVQA |
| TAM-CL | **69.15** | **10.09%** | **11.74%** | **14.60%** | **16.79%** |
| Finetune | 67.27 | 76.18% | 55.89% | 50.07% | 75.11% |
| EWC | 68.43 | 78.72% | 54.84% | 54.04% | 79.46% |
| FDR | 58.73 | 38.38% | 40.97% | 47.51% | 52.12% |
| ER | 67.63 | 16.08% | 12.26% | 24.12% | 19.80% |
| Dytox | 69.01 | 18.20% | 23.63% | 26.89% | 29.90% |

Table 8: Accuracy and forgetting rate of task order: COCOQA → GQA → SNLI-VE → OKVQA → NLVR2