# OpenReview forum: "Task-Attentive Transformer Architecture for Continual Learning of Vision-and-Language Tasks Using Knowledge Distillation"
_EMNLP/2023/Conference — EMNLP 2023 Findings_

### Official Review · Reviewer_94xT · 2023-08-05

**Soundness:** 3

**Excitement:**

3: Ambivalent: It has merits (e.g., it reports state-of-the-art results, the idea is nice), but there are key weaknesses (e.g., it describes incremental work), and it can significantly benefit from another round of revision. However, I won't object to accepting it if my co-reviewers champion it.

**Missing References:**

[1]. De Lange, Matthias, et al. "A continual learning survey: Defying forgetting in classification tasks." IEEE transactions on pattern analysis and machine intelligence 44.7 (2021): 3366-3385.

**Paper Topic And Main Contributions:**

The paper proposes a dynamically expanding transformer-based continual learning architecture for learning multimodal vision-and-language tasks, in addition, the paper applies knowledge distillation and experience reply to further prevent catastrophic forgetting.

**Questions For The Authors:**

1. Does the task attention block include the task-focused attention layer or not? In line 254, the paper says the task attention block takes the task token as input, but in Figure 2 on the right hand side, the task-attention block architecture includes the task attention layer.
2. Task token is generated from task attention layer, how to ensure the same task token is always generated for the same task after training multiple tasks? Is there an evaluation on the consistency of the task tokens?
3. Why to change the task sequence order in ablation study? Results in Table 3 shows that the experience replay module plays the most important role while the task attention block is the least, it makes one wonder whether when evaluating in the same order as Table 2, without TAB, performance can still outperform state-of-the-art methods.

**Reasons To Accept:**

1. Instead of designing a task attention block for each new task, the paper proposes a single task attention block that takes input of task-specific token from a task attention layer to learn task specific knowledge.
2. The paper addresses multimodal vision-and-language tasks.
3. Experimental results show the proposed method outperforms some state-of-the-art methods.

**Reasons To Reject:**

The paper only discusses continual learning for transformers in related work, however, continue learning has been widely explored in the literature for several years, ideas go beyond a specific model architecture. And dynamic expanding architecture and knowledge distillation has been widely explored, the paper fails to claim its novelty compared to previous work. Please refer to reference [1] in the Missing References section, or any other survey paper on continue learning.

**Reproducibility:**

4: Could mostly reproduce the results, but there may be some variation because of sample variance or minor variations in their interpretation of the protocol or method.

**Reviewer Confidence:**

4: Quite sure. I tried to check the important points carefully. It's unlikely, though conceivable, that I missed something that should affect my ratings.

---

> ### Author Rebuttal · Authors · 2023-08-28
>
> Thank you for your time and effort. We are glad that you have confirmed that our results support SOTA performance of our work. We are hopeful that the reviewer is open to engage post-rebuttal discussions so we can address the concern about the novelty of our work.
>
>
> Reasons to reject
>
> We agree that knowledge distillation and dynamic expanding architecture have been explored before. We are not making any contribution claim regarding these ideas either. However, please note that most works on CL are limited to uni-modal scenarios and specifically vision modality. Extending most uni-model CL methods to handle multi-modal tasks is not trivial because most of them use CNN architectures that are not known to be good models for NLP domain. The nice aspect of focusing on a transformer architecutre is that it allows handling multimodal inputs. On the other hand, if we change the base model of DyTox to ViLT and run it to V-L tasks in the comparative experiments, the results are quite unstable. The degraded performance implies that using “dynamic model expantion” and “knowledge distillation” in multi-modal learning is not trivial and further exploration is necessary. TAM-CL is a novel architecture for CL that focuses on learning multi-modal tasks, with the unique combination of dynamic token expansion + intermediate knowledge distillation, which is the innovation of our contribution. This is the main reason that as mentioned by the reviewer leads to SOTA perofrmnace of our method.
>
> Response to the Questions:
>
> 1. Yes, task attention block includes the task attention layer, as shown on the right side of Figure 2. The task token is associated with a specific task, and it is concatenated with the output from previous layer, so it is included in the “embedding” of right side of Figure 2.
>
> 2. When a new task comes in, a task token will be initialized and stored in the task token list. During the training, for the same task, the same task token will be used and modified. After the training of that task, the corresponding task token will be stored. In other words, the task token for a single task is only generated once at the very beginning when we learn that task and will never be regenerated. The same task token, after training and storing, will be applied to that task during the test stage.
>
> 3. This is a great question and touches on one of the less explored aspect of CL in the literature. The reason we changed the task order is expressed in  line 408. We do not have any control on the task order in a CL setting but the tasks are not equally difficult. However, most CL methods only study a single task order which ignores the effect of task order but for different task orders, the performance of CL algorithms may not be equally good. To demonstrate that irrespective of the task order, our methods leads to SOTA performance, we offered experiments on different tasks orders. Hence, we think that these experiments indeed is a strength for our empirical exploration.

---

### Official Review · Reviewer_inPu · 2023-08-10

**Soundness:** 3

**Ethical Concerns:**

Yes

**Excitement:**

2: Mediocre: This paper makes marginal contributions (vs non-contemporaneous work), so I would rather not see it in the conference.

**Missing References:**

N/A

**Paper Topic And Main Contributions:**

This paper aims to address the issue of  continual learning in multimodal data (vision and language) tasks, and the authors claim to be the first to propose a solution for the continual learning problem in multimodal data. To tackle this problem, the authors introduce a teacher-student-based training approach. Experimental results demonstrate that their approach achieves lower forgetting rates.

**Questions For The Authors:**

See above.

If I have any misconceptions about the contributions, please raise them in the rebuttal briefly. Thank you!

**Reasons To Accept:**

- The model achieved state-of-the-art performance in the experiments.
- The paper introduces a continual learning solution for multimodal data, which is the first work in this field.

**Reasons To Reject:**

1. Since the authors claim to present a solution for the CL problem in multimodal data for the first time, they should provide a comprehensive comparison with baselines:

1.1 The comparison with previous work is limited. The compared approach Dytox was proposed in 2022, but the rest approaches are older. There are many CL algorithms proposed recently that should be compared instead of the older ones.

1.2 Apart from the compared methods, the effects of large-scale multimodal models like GPT-4 should be showcased. I know that due to the different baselines, the author's approach might not be directly comparable to large models like GPT-4. However, showcasing such results would help readers grasp the true value of this paper.

1.3 The choice of ViLT model with pre-trained weights (BERT-base-uncased) as a baseline is questionable. Why not combine the CLIP image encoder and language model (like Frozen, MAGMA, etc)? This would provide a clearer assessment of whether the proposed CL method should be applied to the latest architectures and help the readers to judge the values of this approach.

1.4 In the present day, numerous technological paradigms have undergone transformations. Approaches such as LoRa, adapters, and others are evidently capable of addressing the challenges posed by the continual learning (CL) problem. It would be beneficial for the authors to compare their approach against these methods. I recognize that making such a request might appear rigorous, and the authors might face constraints in fulfilling it. It seems that they focus on the CL problem. However, considering that the paper also introduces innovative concepts in model architecture, the contributions might appear somewhat scattered. Consequently, I recommend that the authors consider emphasizing the primary contributions of their work and focusing on a specific aspect (an algorithm for CL problem from the ML perspective).

2. As stated by the authors in the limitation section, their CL approach can also address tasks that are vision-only or language-only. In my view, this indeed seems to be the case. The authors claim that their method is tailored for multi-modal applications, but I did not observe too much aspects of the design specifically intended for multi-modal tasks. If the method can effectively be applied to single-modal data, the authors should produce a solid paper focusing on the CL approach, rather than presenting an engineering submission with limited contributions for several fields and insufficient comparisons.

**Reproducibility:**

4: Could mostly reproduce the results, but there may be some variation because of sample variance or minor variations in their interpretation of the protocol or method.

**Reviewer Confidence:**

3: Pretty sure, but there's a chance I missed something. Although I have a good feel for this area in general, I did not carefully check the paper's details, e.g., the math, experimental design, or novelty.

**Typos Grammar Style And Presentation Improvements:**

Btw, the summary of contributions in the introduction fails to highlight the innovation. Additionally, there is room for improvement in writing to enhance clarity.

---

> ### Author Rebuttal · Authors · 2023-08-29
>
> Thank you for your time and effort. We are glad that you have identified our SOTA performance and the pioneer nature of our work. Given these important aspects, we hope you engage in post-rebuttal discussions and in case we can address your concerns, you reconsider the final rating for our work. We also appreciate that you have mentioned that you are open to discussions during the post-rebuttal period.
>
> 1.1 We appreciate if you can  list the recent proposed CL methods that you have in your mind  in detail so we can add them to our comparisons. We agree that there are many recent methods but due to the fact that most of them are developed for computer vision task, their backbone models are convolutional neural networks instead of transformers. As a result, it is not trivial to extend them to handle multimodal tasks. .
> (Working on adding X-DER into baseline.)
>
> 1.2 Could you please explain how to “showcase” the performance of GPT-4 in our setting by adding more detaisl? Actually, LLMs are not a primary consideration in CL. If you check the CL literature, you can see most continual learning methods (e.g., iCARL, ER, Dytox, GEM, DER, Adapters, etc)  are not comparing their performance with Large-scale models, but only with other CL methods, which use CNN layers, BERT, ViT, and other similar architectures as backbone. As a result, we are unsure how to include GPT-4 in our baseline when it is a model for language tasks. Please also note that GPT-4 is an API-based model, updated frequently from OpenAI’s side, and we do not have the freedom to change it to enable it to handle multimodal tasks.
>
> 1.3 We acknowledge that there are alternatives for ViLT to do our experiments and we could base our exploration on other models. However, the main idea of the experiment is to show that using exactly the same base model, i.e., ViLT, TAM-CL generates better results than other continual learning methods to handle multimodal tasks. We also agree with you that if we change the base mode from ViLT to others such as CLIP, the performance may increase but please note that our computational resources are limited to do the experiments with CLIP in a week. We are running results for CLIP in the background and will update the results if we could generate new results during the post-rebuttal period. However, we respectfully ask the reviewer to focus on the broad message of our experiments that demonstrate TAM-CL outperforms alternative methods.
>
> 1.4 We’ve done the experiment of using adapters as a baseline, as one of the methods mentioned in your comment. Here is the comparison result:
>
> Adapter:  77.50 (0%), 68.89 (1.49%), 30.17 (0.57%), 69.11 (1.28%), 53.88
>
> TAM-CL: 66.09 (13.15%), 66.07 (14.87%), 21.24 (22.59%), 64.05 (19.13%), 50.86
>
> Note that these two lines of comparison follow the same format as Table 2 in the paper. We acknowledge that the Adapter has a leading performance compared with our methods, and its forgetting rate of about 1% is also impressive. However, we don't take it as a regular type of continual learning method due to its high space complexity. For adapters, it inserts 2 adapter modules in each attention layer, l, for every new task, t, along with task-specific head, the space complexity of it is O(l*t) + O(t), while the first term is for the space growth of adapter module and the second term is for the space growth of heads. Such a high space complexity is, to some extent, atypical in the CL method community, so it can easily defeat most of the SOTA CL methods, not only ours, with its high accuracy and low forgetting rate. On the other hand, our method's space complexity is O(1) + O(t), as we have a single-layer task-sharing TAB, and only the heads are task-specific. As a result, we consider methods such as adapters are not comparable.
>
> 2
> Please note that the choice of adding intermediate Knowledge Distillation into the loss objective function is a design specific for multi-modal to fuse the modalities well.. For other similar dynamic-architecture-based methods, such as DyTox, a vision-only approach, the knowledge distillation is applied to the end of the classifiers, not in the middle. We also tried to apply knowledge distillation loss to the output of the classifier when designing the architecture, but the result was much worse, which implies that such an arrangement of knowledge distillation is not suitable for multi-modal situations, and our choice of using intermediate knowledge distillation is the one that fits.
>
> While we agree that adding new experiments based on the reviewer’s feedback will strengthen our empirical results, but we respectfully ask the reviewer to consider that, as stated by the reviewer, this is a “first work” of its own kind and the novelty of the setting and idea can be outweighed when evaluating the work compared to well-established settings where extensive experiments are necessary to warrant a worthy contribution. We hope the reviewer is open to increasing the rating if we address the concerns.

---

### Official Review · Reviewer_c6E7 · 2023-08-11

**Soundness:** 4

**Excitement:**

3: Ambivalent: It has merits (e.g., it reports state-of-the-art results, the idea is nice), but there are key weaknesses (e.g., it describes incremental work), and it can significantly benefit from another round of revision. However, I won't object to accepting it if my co-reviewers champion it.

**Paper Topic And Main Contributions:**

This paper presents a novel method for continual learning of vision-language tasks. The components proposed in this approach result in remarkable continual learning performance.

**Questions For The Authors:**

Please respond to the concerns listed in the weaknesses.

**Reasons To Accept:**

Training a task-specific classifier and task token is a concise and intuitive solution for adapting various tasks. And the teacher model and memory replay effectively promote performance and alleviate catastrophic forgetting.

**Reasons To Reject:**

Weaknesses:

A. The explanation for the expansion of the task-specific classifier is a bit confusing, especially Eq. 8. All tasks use the same classifier with accumulated weights, or each task has its own task-specific classifier?

B. The absolute performance on VQA tasks is weak compared to existing zero-shot vision-language models. It's helpful to validate the performance of TAM-CL on more recent base models, e.g. BLIP2.

**Reproducibility:**

4: Could mostly reproduce the results, but there may be some variation because of sample variance or minor variations in their interpretation of the protocol or method.

**Reviewer Confidence:**

3: Pretty sure, but there's a chance I missed something. Although I have a good feel for this area in general, I did not carefully check the paper's details, e.g., the math, experimental design, or novelty.

---

> ### Author Rebuttal · Authors · 2023-08-28
>
> Thank you for your time and efforts. We are glad that you have found our solution to be effective. We hope you are open to reconsidering our work and giving us a second chance by providing a rating with higher confidence. We have reponed below:
>
> A.Please note that the classifier is task-specific which means that for each of the new tasks, there will be a completely new initialized classifier. However, as the number of tasks increases, the size of new initialized classifier is cumulated. For example, task one’s classifier has an output label size of 100, and task two’s output label size is 50, then task two’s classifier has an output label size = 100 + 50 = 150. This increase is because although we only use intermediate knowledge distillation in our loss objective, we still preserve the choice of adding knowledge distillation to the final prediction as our future work. Due to the special property of visual-language tasks, especially VQA tasks, the output label size varies a lot from task to task. To do knowledge distillation from final output, we should have the two predicted vectors in the same dimension. As a result, with the cumulative classifier, we can use the output of task one’s classifier, with the dimension of [100,1], and the first 100 values of task two’s classifier [:100, 1], to compute the knowledge distillation loss. We hope the above addresses your concern but please let us know if it is not clear.
>
> B. We acknowledge that the absolute performance of VQA tasks may not be state-of-the-art, which is mainly due to the fact that we apply ViLT as our backbone transformer. In TAM-CL, the proposed architecture is shown in the yellow and red blocks of the main diagram, it is a generalizable add-on architecture that can be embedded into most existing V-L transformers, such as the BLIP2. In our comparison experiment, we are showing that with exactly the same backbone transformer, ViLT in our case, we are embedding ViLT with different continual-learning methods, such as EWC, Experience Replay, FDR, Dytox, etc, and TAM-CL is the best among all the methods. We can also change ViLT in all the baseline methods to other transformers, and the absolute performance will definitely improve. However, our purpose is to show that TAM-CL is more helpful, as an add-on continual learning architecture, than other methods relatively. Adding experiments with BLIP2 can be helpful but unfortunately, our GPU resources are limited to do the relevant experiences in only a week. Our hope is that you base your evaluation on the experiments that we have provided.

---

### Meta-Review · Area_Chair_rGNT · 2023-09-15

**Recommendation:** 4

**Metareview:**

Three reviewers provided feedback for this paper. There was a high degree of consensus amongst reviewers in the overall reviews, soundness and excitement. They found the method interesting and effective. They also found the work novel. One reviewer complained about missing baselines but failed to provide the specific ones they had in mind. This reviewer also asked the authors to compare their work to GPT-4 which is unfair in my opinion, given that the multimodal version of GPT-4 isnt even publicly available. Other complaints included a request for more experiments (with BLIP2) and missing related work. Overall the authors have done a good job at addressing these concerns and overall I am satisfied with the paper given the reviews and rebuttal.

---

### Decision · Program_Chairs · 2023-10-07

**Decision:**

Accept-Findings

**Comment:**

Three reviewers provided feedback for this paper. There was a high degree of consensus amongst reviewers in the overall reviews, soundness and excitement. They found the method interesting and effective. They also found the work novel. One reviewer complained about missing baselines but failed to provide the specific ones they had in mind. This reviewer also asked the authors to compare their work to GPT-4 which is unfair in my opinion, given that the multimodal version of GPT-4 isnt even publicly available. Other complaints included a request for more experiments (with BLIP2) and missing related work. Overall the authors have done a good job at addressing these concerns and overall I am satisfied with the paper given the reviews and rebuttal.